# Dynamic and tunable metabolite control for robust minimal-equipment assessment of serum zinc

Monica P. McNerney [1], Cirstyn L. Michel[1], Krishi Kishore [2], Janet Standeven [2] & Mark P. Styczynski [1]*

Bacterial biosensors can enable programmable, selective chemical production, but difficulties incorporating metabolic pathways into complex sensor circuits have limited their development and applications. Here we overcome these challenges and present the development of fast-responding, tunable sensor cells that produce different pigmented metabolites based on extracellular concentrations of zinc (a critical micronutrient). We create a library of dual-input synthetic promoters that decouple cell growth from zinc-specific metabolite production, enabling visible cell coloration within 4 h. Using additional transcriptional and metabolic control methods, we shift the response thresholds by an order of magnitude to measure clinically relevant zinc concentrations. The resulting sensor cells report zinc concentrations in individual donor serum samples; we demonstrate that they can provide results in a minimal-equipment fashion, serving as the basis for a field-deployable assay for zinc deficiency. The presented advances are likely generalizable to the creation of other types of sensors and diagnostics.

---

[1] School of Chemical & Biomolecular Engineering, Georgia Institute of Technology, 311 Ferst Drive NW, Atlanta, GA 30332, USA. [2] Lambert High School, 805 Nichols Rd, Suwanee, GA 30024, USA. *email: mark.styczynski@chbe.gatech.edu

The use of engineered microbes as smart or responsive chemical production systems that produce different outputs in response to different environmental stimuli holds great potential for diverse applications, such as on-demand pharmaceutical production, targeted long-term drug delivery, and minimal-equipment diagnostics. Microbial cells respond to their surroundings, produce complex chemicals, self-replicate, and thrive in complex and harsh environmental conditions, making them excellent candidates to meet the need for responsive, robust, and scalable chemical production. However, the creation of such responsive microbial factories is hindered by the disconnect between biosensor development and microbial metabolite production: most complex cell sensors produce simple reporter proteins in response to target signals[1,2], and most metabolically engineered cells constitutively produce a single target chemical. Incorporating metabolite outputs into microbial sensors brings new difficulties and complications that require new biosensing and metabolic engineering strategies.

Engineering cells to selectively produce only one of multiple possible metabolite outputs (rather than protein outputs) poses many challenges. Expression of multiple enzymatic pathways can be metabolically burdensome[3], which leads to low cell viability and construct stability. Perhaps even more importantly, small amounts of uninduced, baseline enzyme expression can produce large amounts of metabolites at undesired times or conditions, confounding cell output selectivity and preventing a controlled response[4,5]. Metabolic engineering can improve yields and titers of desired metabolites, but the requirement of high output selectivity is not a consideration in typical metabolic engineering strategies[5]. In fact, this selectivity is actually at odds with typical emphases on improving titer, which often lead to leakiness even under repression.

Beyond selective production of target metabolites, sensor cells must also respond to relevant ranges of analyte concentrations. A diversity of effective approaches to shift sensor response points could help facilitate the development of tunable, more applicable biosensors. Most sensors that use bioprospected components respond to concentrations dictated by the natural sensor-analyte affinity, which often fall outside of industrially or clinically relevant response ranges. Currently, the most effective approaches to shift sensor response curves require extensive protein engineering[6] or the use of multiple receptors with varied binding affinities for the target analyte[7,8], which can necessitate further bioprospecting or protein engineering.

As noted above, minimal-equipment diagnostics are one application where environmentally responsive microbial cell factories hold significant promise, via the production of different pigmented metabolites based on the level of some target molecule[9–11]. This application also is a prime testbed for developing approaches to more precisely control metabolite ouput[12–16]. Since pigments are visible to the naked eye, they can be easily used as reporters to assess metabolite control and thus allow faster design-build-test cycles. Further, since pigments are made through multi-enzymatic pathways, approaches used to control pigment production should be applicable to control of other complex metabolic pathways.

Our continued development of a pigment-based bacterial biosensor for assessment of zinc begins to address the challenges associated with harnessing metabolic pathways for biosensing applications[10,11,17,18]. A completed whole-cell zinc sensor could serve as a point-of-care test for zinc deficiency (which is responsible for over 100,000 childhood deaths annually[19]) that overcomes current challenges that prevent diagnosis and treatment in the low-resource areas where zinc deficiency is most prevalent. Currently, no field-deployable test for zinc deficiency exists: current zinc assessment methods require acid digestion of samples and subsequent analysis with mass spectrometers, a process that is costly and labor intensive. Further, to prevent red blood cell lysis, samples must be stored and shipped on ice to analysis laboratories, which introduces additional cost and logistical challenges. A test for zinc that could be performed either on site or in a minimally equipped regional reference lab could dramatically expand the scope of nutritional surveillance and supplementation programs. In some of our previous work, we show that zinc-responsive transcriptional elements could control production of different pigmented metabolites, but this proof-of-principle biosensor is far from field-deployable. Sensor cells require long incubation times (~24 h) even in ideal laboratory conditions, they do not respond to clinically relevant zinc concentrations, test output is difficult to interpret, and the cells are not directly field-deployable. To move this closer to a field-ready test, we sought to redesign the system so that cells (1) produce visible pigments within 4 h of sample addition, (2) respond to clinically relevant serum zinc concentrations, and (3) produce colors that are easy to interpret with minimal equipment.

Here, we describe the development of a robust whole-cell zinc biosensor that accurately quantifies clinically relevant concentrations of zinc in human serum. We design and engineer a library of dual-input promoters that decouple pigment metabolism from cell growth and thus enable visible output within four hours of sample addition in laboratory conditions. We then use transcriptional, translational, post-translational, and metabolic control methods to tune which zinc concentrations activate different pigment pathways, ultimately shifting the threshold zinc concentrations by nearly an order of magnitude compared to our previous proof of principle and into the clinically relevant range. The resulting cells can be used to assay unprocessed serum in a robust, field-friendly fashion: lyophilized cells can be rehydrated with small volumes of human serum, and they produce one of three different visible pigments based on the concentration of serum zinc. The user can assess output qualitatively through visual test assessment or quantitatively with an easy-to-use smartphone app. This work is a major step towards a field-deployable micronutrient biosensor that would impact the treatment of millions of people, and demonstrates generalizable strategies for the development of robust and tunable sensor cells.

## Results

**Synthetic decoupling of chemical production and cell growth.** Our previous efforts to engineer diagnostic zinc sensor cells were plagued by long assay times, construct instability, and poor test interpretability, which needed to be overcome in order to make a potentially deployable diagnostic. To accomplish this goal, we sought to decouple pigment production from cell growth by using a small molecule inducer as a master regulator to control expression of all pigment pathways. During a pre-assay inoculum production stage (in the absence of inducer), cells should be colorless, which should decrease the metabolic burden on cells during this growth phase and thus improve cell viability and genetic circuit construct stability[17,20–23]. A large inoculum of these cells would then be added to the sample, and a standard inducer (such as IPTG or arabinose) would activate the color synthesis system, leaving zinc-responsive elements to control pigment production (Fig. 1a). The assay time would then depend only on the pigment biosynthesis rate, and not the time needed for generation of visible biomass from a small (colored) inoculum, as in our proof-of-principle work. This would also improve test interpretation (since colorless cells will indicate an incomplete or nonfunctional test).

To engineer a system in which pigment production is controlled by both a standard inducer and zinc, we designed

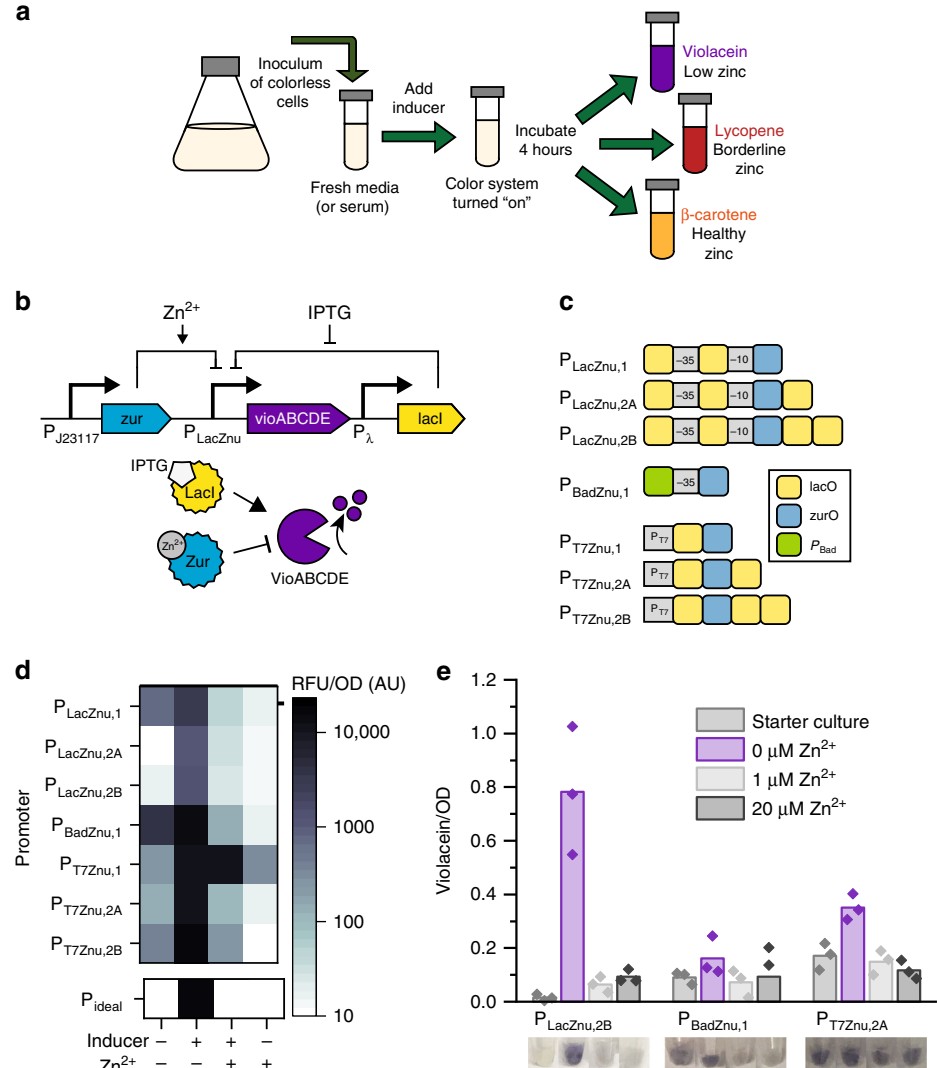

**Fig. 1** Dual-input promoters decouple cell growth and pigment production. **a** Design of a dual-input system that decouples pigment production from cell growth. Colorless cells are added to fresh medium or to a serum sample at a density high enough that cells are visible, and a small-molecule inducer is added to activate the color-response circuit. After a short incubation, different pigments (either violacein, lycopene, or β-carotene) are produced to indicate different zinc concentrations. **b** Circuit diagram and schematic depicting the design of a dual-input promoter that regulates production of the violacein pathway genes, using IPTG/LacI-regulation as an example. Violacein should only be produced in low zinc conditions when IPTG is present. Analogous schematics for AraC- and T7 RNAP- mediated control are in Supplementary Fig. 1. **c** Promoter architecture for the library of synthetic promoters. The −10 and −35 σ70 binding domains are marked, and lac and zur operator sites are shown in yellow and blue, respectively. For the $P_{BadZnu}$ promoter, the −10 binding domain is not explicitly marked because it is embedded in the Zur operator site, and the green box indicates the entire $P_{Bad}$ promoter sequence that is upstream of the −35 binding domain. For $P_{T7}$ variants, the consensus T7 RNAP binding domain is marked in gray. Supplementary Table 1 contains annotated promoter sequences for all constructed promoters. **d** Fluorescent characterization of all engineered promoters. Ideally, eGFP should only be produced in the + inducer/−zinc state, as indicated by example output from a hypothetical ideal promoter ($P_{ideal}$). Each box in the heat map is the average of three biological replicates. Supplementary Fig. 2 shows averages with standard deviations. **e** Violacein production from the best-responding hybrid promoters of each group. A starter culture (−inducer/−zinc) was used to inoculate cultures that contained the appropriate inducer and different concentrations of zinc. Bars represent the average of three biological replicates, which are depicted as overlaying points. Images below the graph show representative cell pellets from each condition. Source data for **d** and **e** are provided in the Source Data file.

dual-input synthetic promoters that contain operator sites for both the inducer's cognate transcriptional regulator (such as LacI) and for the zinc-responsive repressor Zur. During the pre-assay inoculum production stage, LacI (for example) binds to its cognate operator and prevents expression of downstream pigment production genes. IPTG addition should alleviate this repression, with gene expression then solely controlled by Zur, a repressor that binds to its cognate operator in the presence of zinc. Ideally, expression from this dual-input promoter would only occur at very low zinc concentrations (Fig. 1b). The

transcriptional regulators AraC and T7 RNA polymerase have different regulatory mechanisms, but could enact the same logical function (Supplementary Fig. 1).

We created a first generation of dual-input promoters (referred to as $P_{LacZnu,1}$, $P_{BadZnu,1}$, and $P_{T7Znu,1}$) by adding a single Zur operator site downstream of a standard inducible promoter (Fig. 1c, Supplementary Table 1). Addition of operator sites has been successfully used in previous efforts to make promoters controlled by two inputs[24–27], and while fluorescent characterization showed that these first-generation hybrid promoters all

respond to both IPTG and zinc, all promoters show high levels of baseline, uninduced protein expression (Fig. 1d). Zur's operator sequence includes a $-10$ $\sigma_{70}$ RNA polymerase binding domain that is heavily conserved across Zur-regulated promoters[28], so this leakiness is likely caused by unintentional introduction of a promoter element. To decrease this unwanted expression, we engineered a second generation of $P_{Lac}$- and $P_{T7}$-based promoters that include additional LacI operator sites downstream of the Zur operator (Fig. 1c, Supplementary Table 1). The second generation of hybrid promoters all show decreased uninduced expression and improved dynamic range (Fig. 1d).

We determined the best-responding promoter of each class by calculating the relative change in fluorescence caused by inducer and zinc addition (Supplementary Fig. 3), and then used promoters with the optimal dynamic ranges to control production of the metabolic pathway that produces the purple pigment violacein. A starter culture of uninduced cells that had been grown overnight was used to inoculate tubes containing fresh medium, the appropriate inducer, and different concentrations of zinc. Though all systems show some level of induction, the $P_{Lac}$-based system outperforms the others, as it has visually undetectable violacein production in the overnight culture and the highest level of induced violacein production (Fig. 1e).

**A multi-colored fast-responding zinc sensor.** Using the optimized $P_{Lac}$-based violacein circuit, we constructed multi-color zinc-responsive sensor cells by incorporating metabolic pathways from the carotenoid pathway, specifically those to produce the red pigment lycopene and the orange pigment β-carotene. We used LacI/IPTG to control expression of the lycopene production pathway and the zinc-responsive activator ZntR to control expression of the enzyme CrtY, which converts lycopene into β-carotene (Fig. 2a). Upon IPTG induction, cells always produce lycopene, but cells should only appear red at intermediate zinc concentrations: at low zinc concentrations, the more intense pigment violacein overwhelms lycopene and cells appear purple, while at high zinc concentrations CrtY converts all lycopene into β-carotene and cells appear orange. The mevalonate pathway genes (which produce the precursor to the carotenoid pathway) were expressed from $P_{Bad}$ to increase the rate and intensity of carotenoid production.

As designed, the implemented engineered cells turn one of three colors based on the zinc concentration. After overnight growth (with no inducer), cells are visibly colorless and have no detectable violacein or carotenoid. When these cells are used to inoculate medium containing IPTG and different concentrations of zinc, cells appear visibly purple, red, or orange (Fig. 2b) after four hours of culture.

To better enable objective color assessment, we set thresholds for color classifications. We surveyed ten people to ask how they would classify the colors of cells that contained different levels of violacein, lycopene, and β-carotene (Supplementary Fig. 4). Based on these results, cells that contained OD-normalized violacein > 0.2 are considered purple, and cells that contain OD-normalized violacein < 0.2 are classified based on the dominating carotenoid: either red or orange, based on whether cells contain primarily lycopene or primarily β-carotene, respectively.

Though the initial sensor cells respond to a wide variety of zinc concentrations, the concentration range is far wider than the clinically relevant range of zinc concentrations. To meet the requirements for in-field nutrition testing, an assay must use small amounts of serum, require minimal sample processing, and contain enough cells to be visually interpreted. Based on these constraints, we aimed to run 200 μL tests in a 25% serum sample. A simple 1:4 dilution can be easily performed with a fixed-volume

pipette in the field and removes potential inaccuracies caused by minimally-trained personnel in precisely pipetting small sample volumes. A 200 μL test volume is sufficient for cell pellet visualization and requires only 50 μL of serum, a volume that can be reasonably be obtained from a finger-stick of blood[29]. Population data show that serum zinc concentrations fall between 2 and 20 μM[30], so a test run in 25% serum must produce distinct colors across a range of 0.5–5 μM zinc. Across this range, all sensor cells produce primarily lycopene and are classified as red according to predetermined color thresholds. To make a field-deployable sensor, we thus needed to shift both the zinc concentration that triggers the purple-to-red color transition and the concentration that triggers the red-to-orange color transition, overall narrowing the response range by an order of magnitude.

**Rational tuning of color switch points.** We first focused on shifting the purple-to-red response point so that higher zinc concentrations would be required to shut off violacein production. Both colorimetric and fluorescent characterization show that expression of $P_{LacZnu,2B}$ shuts off by 200 nM, and we aimed to shift this by an order of magnitude to a concentration between 1 and 3 μM.

We explored several strategies to increase the threshold concentration that activates the purple to red color transition. We first tested whether reducing the amount of Zur in the system could shift the transition point to higher concentrations. Decreasing Zur expression through ribosomal binding site (RBS) modifications on Zur had little to no effect on Zur's switch point and instead led to increased protein production at all zinc concentrations (Supplementary Fig. 5). We next attempted to use a heterologous zinc-responsive repressor taken from *Bacillus subtilis* (termed $Zur_{Bs}$) to control expression of two *B. subtilis* promoters that demonstrate different in vitro responses to zinc depletion[31]. We placed these operator sites downstream of a standard *E. coli* promoter, and fluorescent characterization shows that $Zur_{Bs}$ fully represses expression from both tested promoter sequences before 1 μM zinc (Supplementary Fig. 6). We then explored protein engineering strategies to create an *E. coli* Zur mutant with a decreased affinity for zinc that maintained a high dynamic range (Supplementary Fig. 7). We mutagenized Zur through both error-prone PCR of the entire gene and through saturation mutagenesis of rationally-selected residues near Zur's zinc-binding pockets and dimerization domains. The best-responding mutants had relatively high GFP production at 1 μM zinc, but had only 2–3 fold differences in expression between 1 and 30 μM zinc (Supplementary Fig. 7), far lower than the nearly 100-fold expression differences between 0 and 1 μM zinc characteristic of wild-type Zur and needed for strong but selective pigment production.

Redesigning regulatory circuit architecture proved much more effective and enabled us to successfully move Zur's effective response point. Instead of altering Zur's activity, we engineered a system such that Zur is only produced (and thus only available for repression) when zinc is present. Building off of a previously described inverter that modestly shifted threshold concentrations for one reporter but was not effective at controlling pigment production[11], we used the zinc-responsive activator ZntR to control Zur expression from the promoter $P_{zntA}$, creating a system that produces more Zur at higher zinc concentrations (Fig. 3a). Thus, even if Zur is effectively all bound to zinc by 0.2 μM and thus in a repressive mode, by modulating the amount of Zur present we can control how much repression of $P_{LacZnu,2B}$ there is. Expression from $P_{LacZnu,2B}$ should thus decrease more slowly with added zinc, and at some threshold zinc concentration,

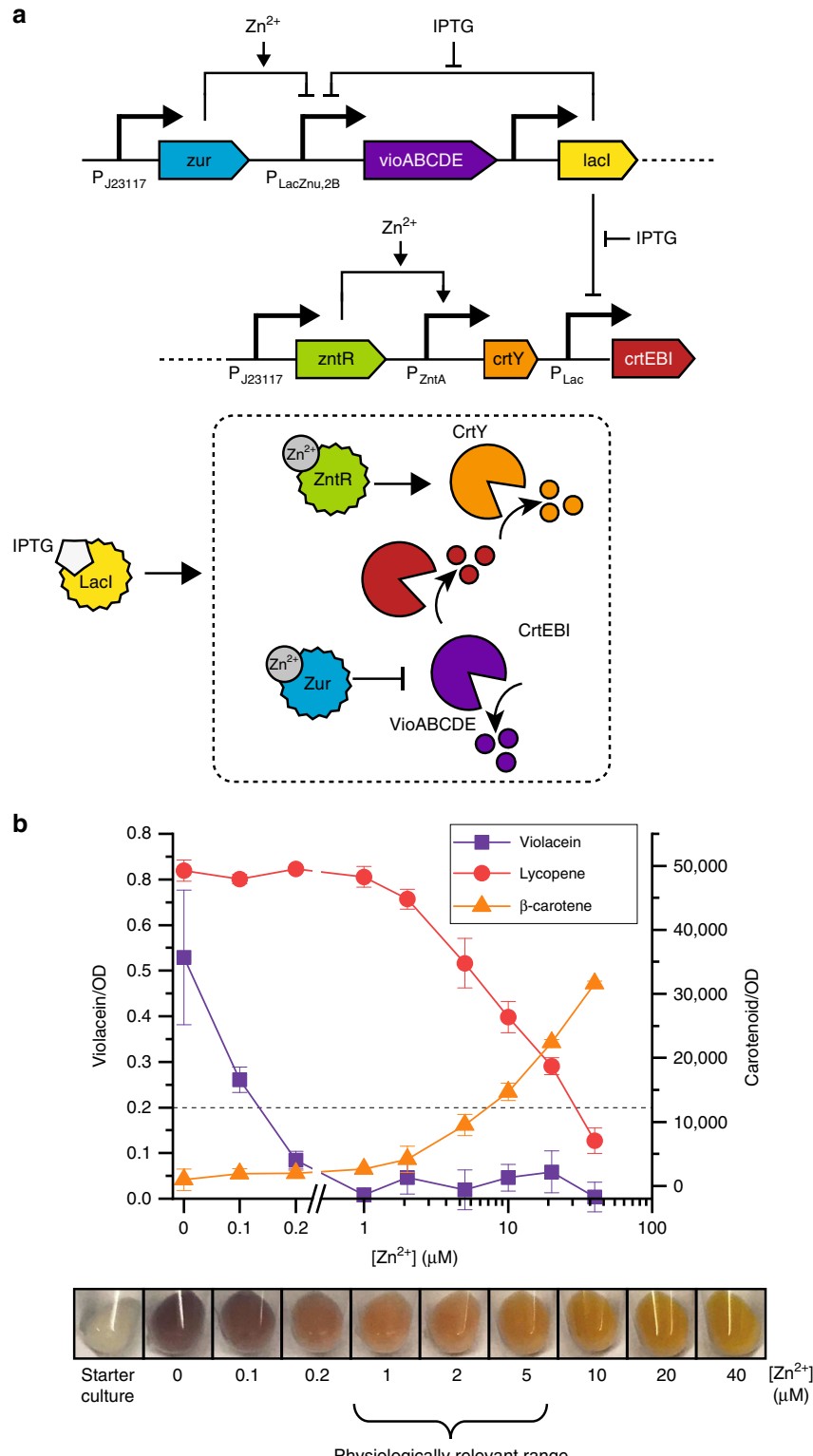

**b**

Figure legend:
- Violacein (squares)
- Lycopene (circles)
- β-carotene (triangles)

Physiologically relevant range

[Zn²⁺] (µM)

expression from $P_{LacZnu,2B}$ should be fully repressed. This threshold zinc concentration can be rationally tuned by varying the RBS and ssrA degradation tag on Zur: systems with low levels of Zur will require more transcription from $P_{ZntA}$ and thus more zinc to shut off violacein expression.

We made a library of sensor cells with this circuit design that turn off expression from $P_{LacZnu,2B}$ at varied zinc concentrations while also maintaining high dynamic ranges. We designed RBSs

with varied strengths using the RBS calculator[32] and used ssrA degradation tags with different relative degradation rates[33] (Supplementary Table 3). Fluorescent protein characterization shows that variation in expression of Zur from $P_{ZntA}$ enables systems that fully repress expression from the $P_{LacZnu,2B}$ promoter at zinc concentrations between 0 and 30 µM (Fig. 3b). Cells show up to 200-fold expression differences between 1 and 20 µM, far higher than the 3-fold changes seen in the best-responding Zur

**Fig. 2** Initial multi-color sensor cells. **a** Circuit diagram and schematic depicting the design of the three-color circuit. During the pre-assay culture stage, LacI represses all pigment production by repressing $P_{LacZnu,2B}$ (which controls violacein production) and $P_{Lac}$ (which controls lycopene production). IPTG alleviates LacI repression, activating the pigment production module indicated by the dashed box in the schematic. The *crtEBI* genes, which control lycopene production, are produced at all zinc concentrations. In low zinc concentrations, violacein is produced, overpowering lycopene production and leading to visibly purple cells. At intermediate zinc concentrations, Zur binds zinc and represses violacein expression, leading to visibly red cells. At high zinc concentrations, ZntR also binds zinc and activates production of CrtY, which converts lycopene to β-carotene, leading to visibly orange cells. **b** Pigment quantification and visualization of sensor cells that were induced with IPTG and grown for four hours. At low zinc concentrations, cells produce violacein and lycopene and appear visibly purple. Cells grown in zinc concentrations between 0.2 and 10 μM are visibly red. At high zinc concentrations, β-carotene production overpowers lycopene production, and cells appear visibly orange. The dotted line indicates the threshold for visible violacein. Error bars indicate standard deviations. The bracket indicates the range of physiologically relevant zinc concentrations corresponding with cells grown in 25% human serum. Ideally, cells should appear three different colors within this bracketed concentration range. Source data for **b** are provided in the Source Data file.

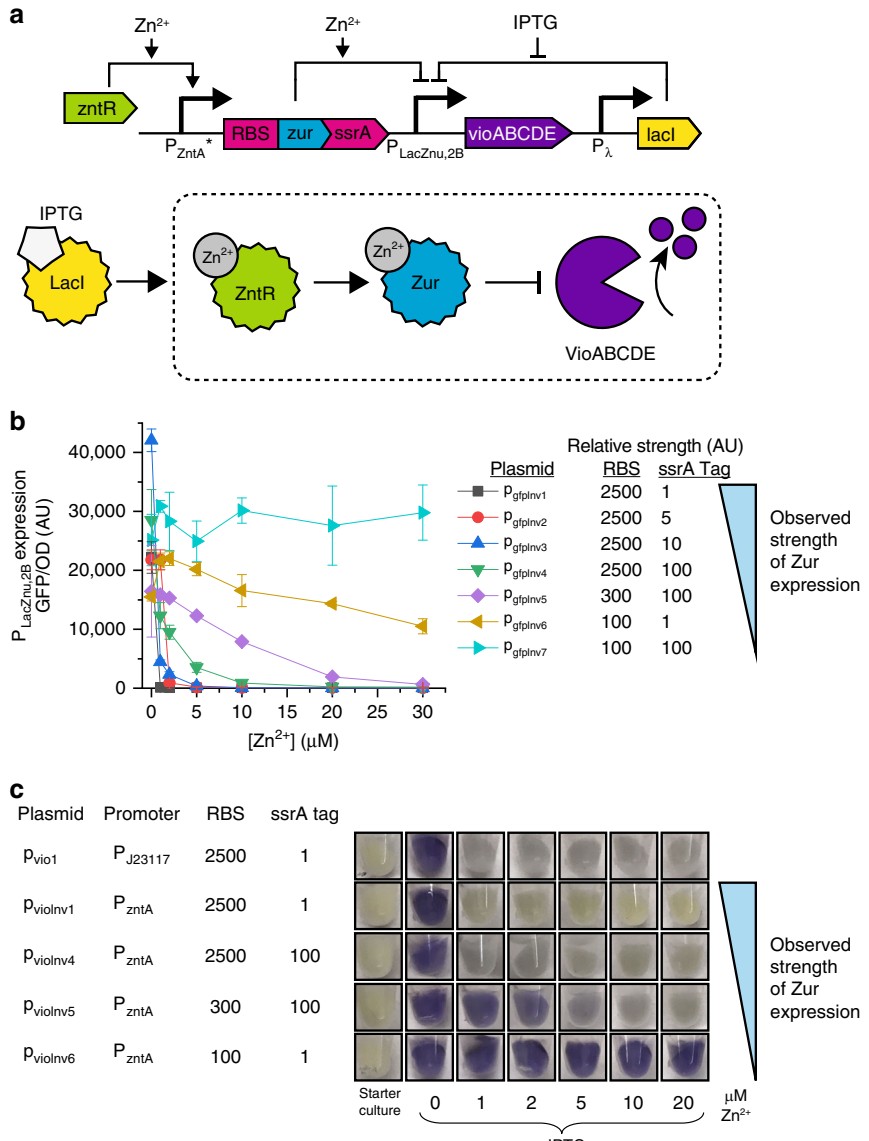

**Fig. 3** Tuning color thresholds with an inverter. **a** Circuit diagram and schematic depicting inverter method to modulate $P_{LacZnu,2B}$ expression. During the pre-assay culture stage, LacI represses all pigment production. Upon IPTG addition, the activator ZntR controls expression of Zur: increasing concentrations of zinc correspond with increased amounts of Zur, which leads to decreasing expression from $P_{LacZnu,2B}$, serving as an effective inverter of the expected output. The RBS and ssrA tag modulate the amount of Zur produced and thus the amount of zinc needed to activate full Zur repression. **b** Fluorescent characterization of Zur inverter circuits. A library of circuits was assembled with different relative levels of Zur. RBS values are the predicted relative RBS strength, and ssrA values indicate the relative strength of degradation (ssrA strength = 1 corresponds with no added degradation tag, and ssrA strength = 100 corresponds with the strongest degradation tag). Decreasing Zur expression levels correspond with cells that shut off expression at higher zinc concentrations. **c** Visual assessment of cells with tunable violacein expression. Overnight starter cultures were added to fresh media that contained IPTG and specified zinc concentrations and grown for four hours. Violacein quantification is shown in Supplementary Fig. 8. Source data for **b** are provided in the Source Data file.

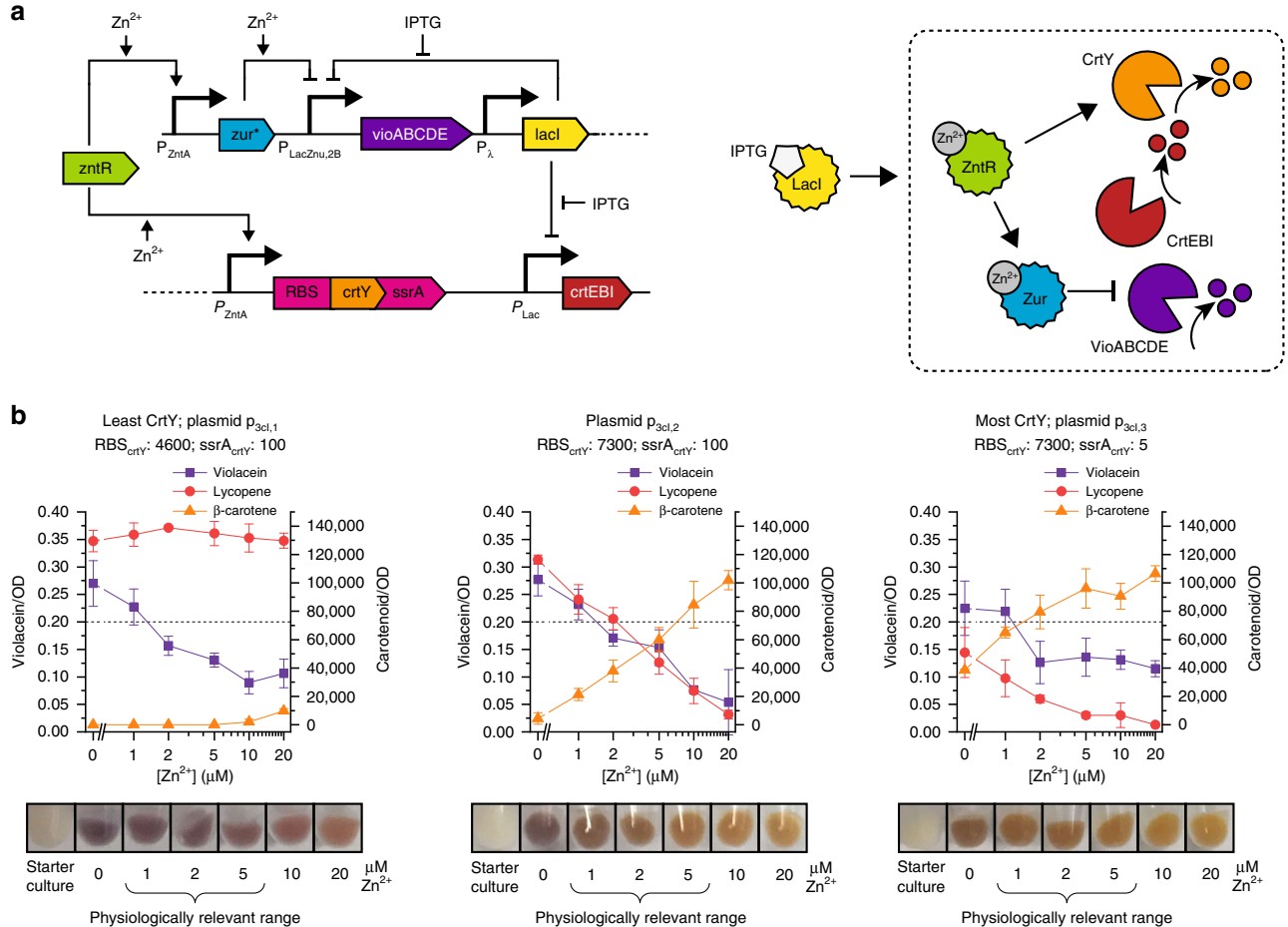

**Fig. 4** Multi-color sensor cells respond to physiological zinc concentrations. **a** Circuit diagram and schematic depicting the design of tunable sensor cells. During the pre-assay culture stage, LacI represses all pigment production. IPTG addition activates the pigment production module, which is indicated by the dashed box. ZntR controls expression of both Zur and CrtY. At a threshold zinc concentration, Zur will repress expression of the violacein pathway, leading to cells that are red at intermediate zinc concentrations. Increasing amounts of zinc also lead to increased amounts of CrtY, which converts lycopene to beta-carotene. Modulation of the RBS and ssrA tag on CrtY can be used to tune the red to orange transition point. **b** Pigment quantification and visualization of sensor cells grown in minimal medium containing glycerol. All sensor cells have violacein off points between 1 and 2 µM, and the lycopene to β-carotene transition points vary based on the expression levels of CrtY. The best performing sensor cells (containing plasmid $p_{3cl,2}$) have moderate CrtY expression and are visibly purple, red, and orange across a physiologically relevant range of zinc concentrations. The dotted line indicates the threshold for visible violacein. Source data for **b** are provided in the Source Data file.

mutants. When these systems control violacein production, threshold zinc concentrations are slightly lower than the threshold concentrations seen in fluorescent characterization, but the off point still ranges between 0 and 5 µM zinc (Fig. 3c, Supplementary Fig. 8). We chose to use the construct that shut off violacein expression between 2 and 5 µM (plasmid $p_{vioInv5}$) for all subsequent assessment.

We incorporated the optimized violacein control circuit ($p_{vioInv5}$) into the three-color expression plasmids, and then tuned the red-to-orange transition point by varying the expression levels of CrtY, the enzyme that converts lycopene to β-carotene (Fig. 4a). This tuning approach is analogous to the approach we used to tune the purple-to-red transition point, but we modulated expression levels of an enzyme, rather than a transcriptional regulator. Cells with lower expression levels of CrtY require a higher concentration of zinc to produce sufficient CrtY to mediate the red to orange color change and thus should appear red over a larger range of zinc concentrations.

Through these tuning approaches, we created a suite of three-color sensor cells with tunable zinc thresholds. We initially performed tests in media that contained glucose as the carbon source (Supplementary Fig. 9), and in these cells, violacein largely

overpowered carotenoid expression and led to murky colored intermediates. When we instead used glycerol as the carbon source, cells show reduced violacein expression and increased carotenoid expression. They then appear distinctly purple, red, and orange (Fig. 4b), and have purple off points between 1 and 2 µM zinc. Cells with the lowest CrtY expression (containing plasmid $p_{3cI,1}$) do not turn orange over the range of zinc tested, and cells with the highest CrtY expression (containing plasmid $p_{3cI,3}$) transition from red to orange between 0 and 1 µM zinc. The best responding cells, which have intermediate CrtY expression (containing plasmid $p_{3cI,2}$), turn from red to orange at ~5 µM zinc, enabling sensor cells that meet the defined criteria of producing three different colors between 0.5 and 5 µM zinc.

**Quantifying relevant zinc concentrations in human serum.** We next used these optimized sensor cells to assess zinc concentration in human serum. Since the transcription factor ZntR controls both violacein and β-carotene production in the circuits that contain the inverter, we confirmed that ZntR specifically responds to zinc, and not to other divalent cations present in serum (Supplementary Fig. 10). Then, we used an inducible lycopene

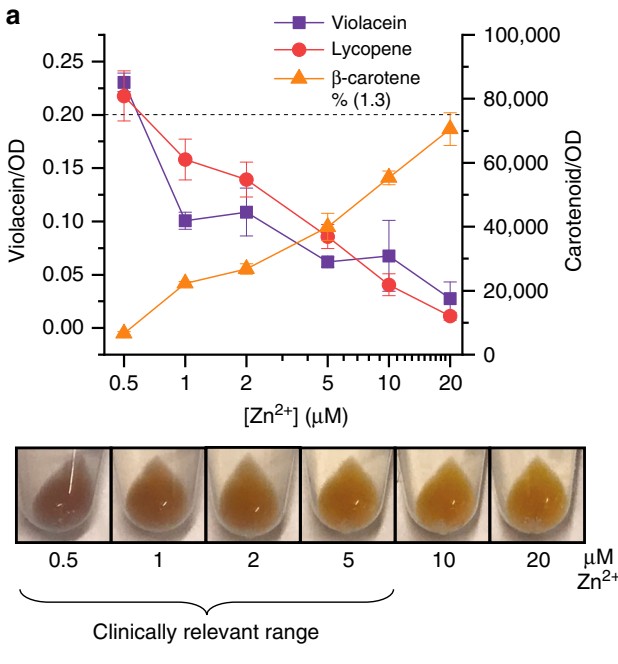

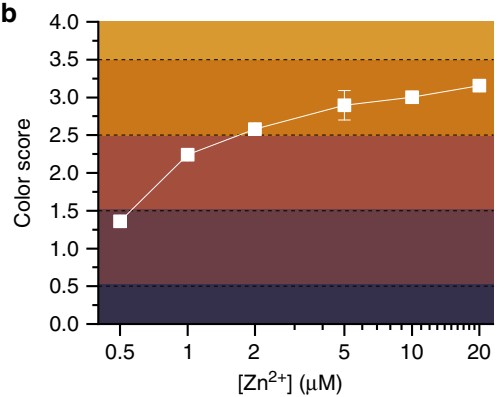

**Fig. 5** Assessment of zinc concentration in human serum. **a** Pigment quantification and visualization of sensor cells grow in 25% serum with IPTG and specified zinc concentrations. Assessment was performed four hours after inoculation. Cells are visibly purple, red, or orange across a range of physiologically relevant zinc concentrations. The dotted line indicates the threshold for visible violacein. **b** Quantitative color assessment of sensor cells. Using RGB values taken from images of cell pellets, a color score was calculated for cells grown in each zinc concentration. As indicated by shading on the plot, scores between 0 and 1.5 indicate purple cells, scores between 1.5 and 2.5 indicate red cells, and scores between 2.5 and 4 indicate orange cells. By RGB score quantification, cells are classified as either purple, red, or orange across a range of 0.5 to 5 μM zinc. Data points show the average of three biological replicates, and error bars indicate standard deviation. Source data for **a** and **b** are provided in the Source Data file.

circuit to show that cells are metabolically active in unprocessed serum as long as they are inoculated to a sufficiently high OD (Supplementary Fig. 11), which is consistent with previous work demonstrating that large inocula are required to enable bacterial growth in human serum[11]. We inoculated a very high concentration ($OD_{initial} = 3.0$) of the sensor cells containing the plasmid $p_{3cI,2}$ into media containing 25% of untreated human serum that had different zinc concentrations. Cells produced different pigments based on the concentration of zinc in the sample, turning either purple, red, or orange across a physiologically relevant range of zinc concentrations (Fig. 5a).

Though our ideal goal is equipment-free test interpretation, cell color results can also be assessed quantitatively, which would eliminate potential subjectivity and variability and could enable more precise and reliable assessment of serum zinc values. Using the RGB values from the color bar used in the survey to assess color thresholds (Supplementary Fig. 4), we developed relationships between RGB values and perceived color. We then quantified the RGB values of photos of cell pellets and calculated a color score that indicates the cell's relative color (with purple = 0–1.5, red = 1.5–2.5, and orange = 2.5–4) (Supplementary Fig. 12). Color scores increase with increasing zinc concentrations, and based on set thresholds, cells grown in the physiologically relevant concentration range of 0.5–5 μM zinc are either purple, red, or orange (Fig. 5b).

**Demonstrating field deployability of a zinc assay.** Upon demonstrating the ability of sensor cells to assess serum zinc in a lab setting, we aimed to show that the assays could be performed in minimally equipped settings. We envisioned compiling a simple kit with supplies to take a finger-stick of blood, tubes of lyophilized cells, a fixed-volume pipette, a previously demonstrated hand-powered centrifuge[34], and a smart phone with an app for test assessment. With this kit in hand, a minimally-trained field worker could isolate serum from a finger-stick of blood and use it to rehydrate sensor cells. After incubating the cells with just body heat, the field worker could centrifuge the cells with the hand-powered centrifuge and assess zinc concentration with a smartphone app (Fig. 6a). To make such a test possible, a few key criteria must be met: (1) sensor cells must be lyophilized for long-term storage at ambient temperature, (2) tests must function in volumes of serum that could be isolated from a finger stick of blood (typically less than 70 μL), (3) tests must work in serum isolated from arbitrary individual donors (and thus not susceptible to matrix effects), (4) test output must be robust to variation in temperature and incubation condition, and (5) a smartphone app must reliably interpret test output.

To create tests that meet these requirements, we first showed that sensor cells function following lyophilization and rehydration in human serum. Using a 10% sucrose solution as a lyoprotectant, we lyophilized cells, rehydrated them in media containing different zinc concentrations, and showed that the final coloration is nearly identical to that of control cells that had been stored at 4 °C overnight instead of lyophilized (Supplementary Fig. 13A). When rehydrated in medium that contains 25% serum, lyophilized cells also produce different colors based on serum zinc concentration (Supplementary Fig. 13B), though the time to coloration is notably longer (~12 h). We then evaluated color production of reactions that contain volumes of serum that could be obtained from a finger stick (~50 μL) and that were incubated with just body heat and agitation (rather than in a shaking incubator in a laboratory environment). Reactions run in these field-like conditions produce color nearly identical to those run in ideal laboratory conditions (Supplementary Fig. 13B), demonstrating the potential for zinc assessment without refrigerated sample storage and incubation equipment.

Using lyophilized cells and body incubation, we next evaluated whether sensor cells could reliably assess zinc levels in serum that we isolated from individual donor blood samples. Because all donors had healthy or borderline zinc levels (Supplementary Table 2), we treated each with a Chelex-100 resin to deplete zinc and then supplemented zinc to create serum with a range of zinc concentrations. In all samples, lyophilized sensor cells produce different colors based on zinc concentration, but the relative color and color intensity varied across experiments, which

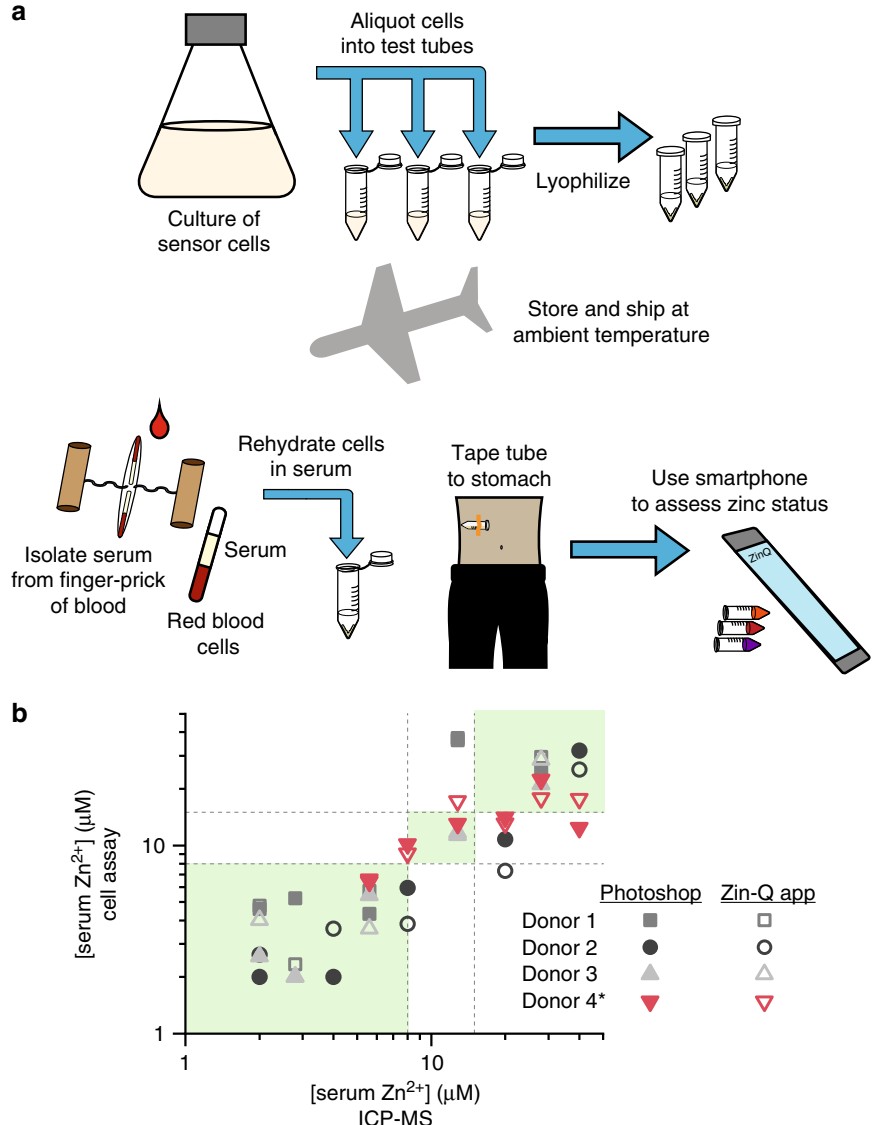

**Fig. 6** Accurate zinc assessment in a minimal-equipment setting. **a** Schematic of proposed workflow for in-field zinc assessment with sensor cells. During test manufacture, a large culture of repressed sensor cells is aliquoted into individual test tubes, and cells are lyophilized. Lyophilized tests can be shipped at ambient temperature to the site of testing, where serum will be isolated from a finger stick of blood and used to rehydrate the lyophilized sensor cells. Sensor cells can be incubated at ~37 °C by taping tubes to the body. After incubation, the test can be interpreted with an easy-to-use smartphone app. **b** Assessment of zinc status from color quantification. Images containing pellets of standards and a test reaction were processed either with Photoshop or with the Zin-Q smartphone app. A calibration curve was calculated from each image, and this was used to assess the concentration of zinc in the test reaction. When results are processed both with Photoshop and with the Zin-Q app, the test accurately classifies all serum samples with low zinc levels. Accurate classification of low, borderline, and high serum zinc is indicated by points in the green area of the plot. The asterisk indicates that serum from Donor 4 was not used in test standards. Source data for **b** are provided in the Source Data file.

we attributed to slight differences in body temperature and motion between tests done on different days.

To enable test interpretation that accounts for this variability, we developed a standardization approach in which serum samples with known zinc concentrations are run as standards in parallel with test reactions that have unknown zinc concentrations. A photo of all six reactions is taken in an inexpensive and lightweight light-controlled setting (see Methods), and the average color of each pellet can be determined via computer software (e.g., Photoshop). The RGB values of the standards are used to make customized calibration curves that are then used to evaluate the zinc concentration of the test reaction (Supplementary Fig. 14). Standards were made by pooling serum from three

donors and adding specified zinc concentrations. When these standards are run in parallel with reactions that contain individual donor serum, the test accurately classifies zinc concentrations in serum taken from all donors, including Donor 4, whose serum was not added to the pool for the test standards (Fig. 6b). Though the test misclassified a few higher zinc concentrations, for the critical regions of low zinc levels the assay classified samples perfectly, and there were very few misclassifications overall.

Finally, we developed a smartphone app that enables all color assessment to be easily performed at the site of testing. A photo can be loaded directly into the app, and the app prompts the user to select tubes that contain each of the standards and the test

reaction. The app processes the color of each cell pellet, creates customized calibration curves for the image, and reports the zinc concentration, classifying it as either low, borderline, or high based on clinically relevant thresholds (Supplementary Fig. 15). Despite variation introduced based on the pixel that the user taps as the center of the pellet, the app reproduces results obtained through manual pellet analysis via image processing software such as Photoshop (Fig. 6b, Supplementary Fig. 16), demonstrating that this zinc assay can be objectively interpreted without any advanced equipment, expertize, or processing.

## Discussion

We have demonstrated the development of a field-deployable pigment-based biosensor for micronutrient testing and have developed generalizable strategies to decrease the time to response, increase cell stability, tune the response range, and account for environmental variability. The final sensor cells turn purple, red, or orange to indicate whether a serum sample has dangerously low, borderline, or healthy levels of serum zinc, and results can be interpreted through visual inspection or with a smartphone app. Compared to previous proof-of-principle work, we reduce assay time, tune the sensor response point by an order of magnitude to clinically relevant concentrations, and show that the test can be reliably and accurately used as the basis for a minimal-equipment assay.

The developed sensor cells can serve as the foundation of a zinc diagnostic that meets all requirements for in-field micronutrient assessment. Tests can be stably stored at ambient temperature, require minimal sample processing, and produce easily interpretable results. If incorporated into standard health surveys, this test could provide governments and aid organizations with the information needed to efficiently implement zinc supplementation programs.

More broadly, the sensor cells we created can serve as a platform for the development of other sensor cells that produce chemical outputs. In the final sensor cells, a single transcription factor mediates multiple metabolite production pathways: ZntR directly mediates production of β-carotene and indirectly mediates production of violacein by controlling expression of Zur, which is effectively a constitutive repressor. By replacing $P_{ZntA}$ with a different metabolite-responsive promoter, one could easily make a pigment-based biosensor for a different target metabolite. *E. coli* has a host of metabolite-responsive promoters[27], enabling easy production of sensors for a range of metabolites. Transcription factors and regulatory pathways from other organisms can also be expressed in *E. coli*, further broadening the scope of biosensor development. Additionally, Zur and its corresponding operator site could potentially be replaced with a more well-characterized transcriptional regulatory system, which could potentially improve certain aspects of the sensor's response. For applications outside of pigment-based biosensing, other metabolic pathways could replace the pigment expression pathways to enable production of a desired chemical upon sensing a threshold concentration analyte.

Recently developed prototyping tools can help facilitate easy incorporation of new metabolic pathways into this sensing framework. Cell-free expression (CFE) systems, which use bacterial protein extract to implement genetic networks, can be used to rapidly prototype metabolic pathways to determine the optimal relative expression levels of each pathway enzyme[35,36]. While CFE systems have garnered attention for use as biosensors[37,38], for applications that require sustained and scalable chemical production, living cells are a more advantageous chassis while CFE systems serve a key function as tools to rapidly explore the design space.

While we were able to effectively shift Zur's response point, our tuning strategy sacrificed Zur's original sharp, step-like response (which would have enabled more discrete color changes) for a linear response over the concentration range of interest. Incorporation of positive feedback loops and more extensive protein engineering efforts could be used to create systems that have both shifted response thresholds and cooperative, switch-like responses. Further optimization could also involve redesigning the system to allow more orthogonal control of specific protein expression levels. To alter levels of both Zur and CrtY, we used protein degradation tags because of their effectiveness in decreasing protein levels. However, using degradation tags to control multiple regulator proteins could introduce competition for proteases and thus inadvertently couple expression levels of the two proteins. Alternative approaches for more precise, orthogonal tuning could rely solely on modulating RBS strength.

As the applications of biosensing expand to include not just sensors, but living therapies, engineering cells to deliver diverse types of payloads upon sensing target molecules will be critical. Responsive bacterial therapies have already been developed for treatment of cancer[39,40], metabolic disorders[41,42], and inflammation[43], and cells that produce complex chemicals upon target recognition could have an even greater impact on disease treatment. As we learn more about the microbiome's role in disease regulation, more potential therapeutic targets for engineered cell therapies will likely arise, expanding the scope of bacterial therapies to entirely new diseases. The presented approaches for incorporating chemical production into biosensors in a robust and tunable fashion can help to enable a new generation of bacterial sensors to meet the need for responsive chemical delivery.

## Materials and methods

**Materials.** T4 DNA ligase, T5 exonuclease, Taq ligase, Phusion polymerase, Q5 polymerase, and restriction endonucleases were purchased from New England Biolabs (Ipswich, MA, USA). E.N.Z.A. Plasmid Mini Kits were purchased from Omega Bio-tek (Norcross, GA, USA), and QIAquick PCR Purification Kits and QIAquick Gel Extraction Kits were purchased from QIAGEN (Valencia, CA, USA). Lycopene (98%) was purchased from Cayman Chemical (Ann Arbor, MI, USA). Sudan I (95%) was purchased from TCI America (Portland, OR, USA).

**Strains and plasmids.** *Escherichia coli* K-12 DH10B (New England Biolabs, Ipswich, MA) was used for plasmid assembly. Lambda red recombination[44] was used to make a DH10BΔ*zur* strain, which was used for all protein and metabolite production. Briefly, the kanamycin resistance cassette was amplified from the plasmid pKD4 using primers that contained the P1 and P2 priming sequences specified by Datsenko and Wanner[44] and an additional 50 nucleotides of E. coli genomic sequence either upstream (P1) or downstream (P2) of the gene for Zur. The polymerase chain reaction (PCR) product was gel-purified and transformed via electroporation into cells that contained the plasmid pDK46, which expresses the lambda red recombinase under control of the PBad promoter. Successful knockouts were selected on kanamycin plates, and replacement of the target gene with the kanamycin cassette was confirmed via PCR. The kanamycin selection marker was not excised. The constructed DH10BΔ*zur* strain was used for all experiments that assessed response to zinc.

The plasmid pSB3T5, with a p15A origin, was taken from the Standard Registry of Biological Parts and used as the backbone vector for all plasmids used in the final analysis. The plasmid pSB6A1, which is a derivative of the pBR322 plasmid, was obtained from the Standard Registry of Biological Parts and used for expression of the mevalonate pathway genes.

**Cloning and construct assembly.** All constructs were assembled with either Gibson assembly[45] or restriction endonuclease digestion of components and subsequent ligation and transformation following the BioBricks idempotent standard assembly[46]. LB medium composed of 10 g L$^{-1}$ NaCl, 5 g L$^{-1}$ yeast extract, and 10 g L$^{-1}$ tryptone was used for all cell growth and cloning steps. The following antibiotics were used for appropriate selection: tetracycline (15 μg mL$^{-1}$), chloramphenicol (34 μg mL$^{-1}$), kanamycin (30 μg mL$^{-1}$), and carbenicillin (100 μg mL$^{-1}$).

The genes *zur* and *zntR* were isolated from previously assembled zinc-responsive plasmids[5]. The coding sequence for Zur from *B. subtilus* (*zur$_{Bs}$*) was codon-optimized for expression in *E. coli*, and the gene was synthesized by Eurofins (Louisville, KY). The lycopene expression cassette (*crtEBI*) was amplified

from a previously assembled vector[17] that uses the genes from part bba_k274100 of the Registry of Standard Biological Parts. The violacein genes *vioA*, *vioB*, *vioC*, *vioD*, and *vioE* were amplified from Part bba_k274002 of the Registry of Standard Biological Parts. A previously assembled arabinose-inducible plasmid expressing the mevalonate pathway[17] used genes from the plasmid pJBEI-6409[47], which was obtained from Addgene (Cambridge, MA, USA). Supplementary Tables 4 and 5 contain sequences of all parts used in this study, and Supplementary Table 6 contains all oligos used in plasmid assembly.

**Cell culture.** For all protein and pigment expression experiments, a modified M9 medium was used. A 5× salts solution consisted of 22.5 g L$^{-1}$ NaCl, 8.2 g L$^{-1}$ KCl, 5 g L$^{-1}$ NH$_4$Cl, 19.5 g L$^{-1}$ MES, 10 g L$^{-1}$ β-glycerophosphate and was pH adjusted to 7.4 with 2 M KOH. To remove trace amounts of zinc, each 1 L batch of salts was treated with 1 g of Chelex-100 resin. The mixture was vigorously stirred for 1 h, and resin was removed through filtration with a 0.2 μm filter. The final medium consisted of 1× salts, 2 mM MgSO$_4$, 0.1 mM CaCl$_2$, 0.01% thiamine, 1.92 g L$^{-1}$ of SC-Ura amino acid mixture (Sunrise Science), and 0.4% of either glucose or glycerol. Medium was sterilized through filtration with a 0.2 μm filter.

For GFP expression, freshly transformed DH10BΔ*zur* colonies were inoculated in triplicate into 4 mL of modified M9 medium and grown at 37 °C and 220 rpm for 24 h. For pigment expression in the absence of serum, freshly transformed DH10B colonies were inoculated in triplicate into 5 mL of modified M9 medium containing the appropriate antibiotic, and grown at 37 °C and 220 rpm for 18 h. Cells were concentrated through centrifugation and used to inoculate cultures that contained fresh medium with the appropriate inducer and zinc concentration. Each 4 mL culture was inoculated to an OD of 0.1 and grown at 37 °C and 220 rpm for 4 h. OD, carotenoid content, and violacein content were measured at 4 h.

For pigment expression in serum, freshly transformed DH10BΔ*zur* colonies were inoculated in triplicate into 3 mL of modified M9 medium containing the appropriate antibiotic and grown at 37 °C and 220 rpm for 8 h. This culture was used to inoculate a 300 mL culture of modified M9 medium containing the appropriate antibiotic, and the 300 mL culture was grown at 37 °C and 220 rpm for 18 h. Cells were concentrated through centrifugation and used to inoculate cultures that consisted of 25% serum, 75% modified M9, and the appropriate antibiotics, inducers, and supplemented zinc. Unless otherwise specified, 1 mL cultures were inoculated to an OD of 3.0 and grown at 37 °C and 220 rpm for 4 h. OD, carotenoid content, and violacein content were measured at 4 h.

**Lyophilization of sensor cells.** To prepare cells for lyophilization, saturated 300 mL cultures of sensor cells were centrifuged, the supernatant was removed, and cells were resuspended in deionized, nuclease-free water to an OD of 3.0. Cells were recentrifuged, the supernatant was removed, and cells were resuspended in an identical volume of a 10% sucrose solution. 200 μL of the solution was aliquoted into 2 mL microcentrifuge tubes, and a needle was used to poke a hole in the lid. Tubes were flash-frozen in liquid nitrogen and transferred to a pre-chilled Labconco Fast-freeze flask that contained a small amount of liquid nitrogen. Care was taken to transfer samples quickly to prevent thawing. Flasks were connected to a Labconco benchtop freeze-drier and lyophilized at −50 °C and 0.05 mbar for 12 h. Tubes were then removed and immediately recapped with a new lid. Tubes were stored in a sealed bag at room temperature until testing.

**Field-friendly conditions.** In experiments that mimic the conditions of a field test, lyophilized sensor cells were rehydrated in 200 μL of modified M9 medium that consisted of 25% serum with the specified concentration of zinc, 75% modified M9, and the appropriate antibiotics and inducers. Microcentrifuge tubes were sealed and taped to the user's stomach with standard lab tape. After 12 h of incubation, cultures were transferred to 0.6 mL microcentrifuge tubes, and tubes were centrifuged at 3000 rcf for four minutes to pellet cells. Tubes were taped on white paper, and a 3D-printed dome with an aperture at the top was placed over the tubes for light control. A smartphone was used to take a picture of the cell pellets. For assessment of individual donor serum (Fig. 6), lyophilized cells were stored at ambient temperature for five days prior to use to model likely transport and deployment conditions.

**Serum processing and zinc measurement.** Pooled human serum was purchased from Corning (Corning, NY). Zinc was removed from serum through Chelex-100 treatment. In total 1 g of Chelex 100 resin was added to 100 mL of serum, and the mixture was vigorously stirred for 2 h.

In single donor experiments, blood was collected from donors as approved in IRB protocol number H17489. Venous blood was collected in 6 mL BD Vacutainer collection tubes for trace element testing, and tubes were left on ice for 30 min to clot. Blood was then transferred to a 50 mL conical tube and centrifuged at 2700 rcf, 4 °C for 30 min. The serum was removed and either immediately frozen or treated with Chelex-100 resin. In total 80 mg of resin was added to 8 mL of serum, and the mixture was vigorously stirred for 2 h. Resin was isolated from the samples through centrifugation, and serum was syringe filtered. All serum samples were aliquoted to minimize freeze-thaw cycles and stored at −20 °C.

For all serum samples, zinc concentration was measured at the University of Georgia Laboratory for Environmental Analysis. Samples were digested with concentrated acid and analyzed on an ICP-MS according to EPA method 3052.

**Measurement of optical density and fluorescence.** Optical density of samples was quantified by measuring the absorbance at 600 nm either in a ThermoFisher Genesys 20 spectrophotometer with a 10 mm path length or in a Biotek Synergy H4 plate reader. When necessary, cultures were diluted to ensure that the reading fell within the linear range of the instruments. 1 mL of culture was used for spectrophotometer measurements, and 150 μL of culture in a 96 well plate was used for plate reader measurements. Calibration curves were made for each instrument and used for uniform reporting of optical density. All optical densities reported correspond with those taken in the spectrophotometer.

For GFP quantification, fluorescence at 485 nm excitation and 510 nm emission were measured on a Biotek Synergy H4 plate reader. All reported fluorescence values are background-subtracted and normalized to optical density.

**Pigment extraction and quantification.** For carotenoid analysis, either 1 mL or 250 μL of bacterial culture was pelleted at 18,000 rcf for 5 min. Cell pellets were resuspended in 50 μL of ultrapure water. Carotenoids were extracted with 1 mL of acetone at 50 °C for 15 min. A heat block was used to maintain temperature and extractions were vortexed every 5 min. Cellular debris was pelleted at 18,000 rcf for 5 min, and 500 μL of supernatant was removed for analysis. Carotenoid content was analyzed on a Shimadzu Prominence UFLC using an Agilent C18 4.6 mm × 50 mm column with a 5 μm particle size and a Shimadzu photodiode array detector. A solvent ratio of 50:30:20 acetonitrile:methanol:isopropanol was used as the mobile phase[13] and run at a flow rate of 1 mL min$^{-1}$ with a 25 μL sample injection volume. Absorption was detected at 471 nm. Retention times and peak intensities were compared to analytical standards spiked into control extractions from DH10B cells.

For violacein analysis, either 1 mL or 300 μL of bacterial culture was pelleted at 18,000 rcf for 5 min. Cell pellets were resuspended in 50 μL of ultrapure water. Violacein was extracted with 40 μL of water-saturated butanol. The mixtures were vortexed for 5 min at room temperature and then centrifuged at 18,000 rcf for 5 min. The butanol layer was removed and recentrifuged at 18,000 rcf for 5 min. 15 μL of the butanol extraction was removed for analysis and added to a 384 well plate. Absorbance at 585 nm was measured on a Synergy H4 plate reader. All reported values are background subtracted, using a butanol extraction of non-engineered *E. coli* cells as the blank.

**Visualization of cell color.** In experiments without serum (Figs. 1–4), 1.67 mL of culture was removed from each replicate and pooled together to make 5 mL of culture for each condition. Cells were pelleted and resuspended in 250 μL of PBS. The suspension was transferred to in 0.6 mL microcentrifuge tubes and centrifuged at 3500 rcf for 5 min. Tubes were taped to white paper, and photos of tubes were taken with an iPhone 7 smartphone.

In experiments with pooled human serum (Fig. 5), 500 μL of culture was transferred to 0.6 mL microcentrifuge tubes and centrifuged at 3500 rcf for 5 min. Tubes were taped to white paper, and photos of tubes were taken with an iPhone 7 smartphone.

In experiments with individual donor serum, each culture was transferred to a 0.6 mL microcentrifuge tube and centrifuged at 3500 rcf for 5 min. Tubes were placed on the ZinQ template according to the marked labels, and an image was taken in a light controlled setting with a Nokia 6.1 smartphone.

**Protein mutagenesis experiments.** For undirected mutagenesis of *zur*, error-prone PCR was used to introduce mutations at a target rate of 2 amino acids per coding sequence using a variation of the protocol described in Wilson and Keefe[48]. Reactions contained 1 mM each of dATP, dTTP, dCTP, and dGTP, 5 μM of the appropriate forward and reverser primer (Supplementary Table 6), 2 mM MgCl$_2$, 0.25 mM MnCl$_2$, 1× Standard Taq Buffer (NEB), and 1.25 U μL$^{-1}$ of Taq DNA Polymerase (NEB). 15 cycles were run with a denaturation step of 30 s at 95 °C, an annealing step of 60 s at 51 °C, and an extension step of 5 min at 68 °C. Mutated sequences were inserted into a GFP expression plasmid through Gibson assembly, and plasmids were transformed into DH10BΔ*zur* cells and grown on modified M9-agar plates with 1 μM zinc at 37 C. After 24 h, plates were visually inspected, and green colonies were streaked onto modified M9-agar plates containing 20 μM zinc and incubated at 37 °C. After 18 h, plates were analyzed, and cells that were visibly green on 1 μM plates and visibly colorless on 20 μM plates were inoculated. Plasmids were isolated from these cells, sequenced, and used for a more thorough analysis of zinc response.

For site-directed mutagenesis of Zur, eight residues were chosen for saturation mutagenesis based on their proximity to the zinc-binding residues or role in the dimer interface. Using primers for NNK mutagenesis (Eurofins) of the specified residue, a GFP-expression plasmid was assembled, transformed into DH10BΔ*zur* cells, plated on modified M9 plates, and grown at 37 °C for 18 h. In total 96 individual colonies were inoculated into 150 μl of modified M9 containing 1 μM zinc in a 96 well plate and grown at 180 rpm and 37 °C. After 12 h, fluorescence and OD were measured, and 7.5 μl of each culture was added to 140 μl of modified M9 media containing 20 μM zinc. After 6 h, GFP and OD were measured.

**Smartphone app for color assessment**. Android Studio was used to create an app to process photos that contain six cell pellets: five standard reactions and one test reaction. The app prompts the user to select each of the six different cell pellets. Following selection of each pellet, the app displays a 200 × 200 pixel area surrounding the initially chosen pixel and prompts the user to re-select the center of the pellet to enable more precise identification of the pellet center. The app averages the RGB values of a 20 × 20 pixel area surrounding the user's second selection and creates calibration curves for each color channel. The zinc concentration of the sample is then calculated from these calibration curves.

**Reporting summary**. Further information on research design is available in the Nature Research Reporting Summary linked to this article.

## Code availablilty

The files containing the code for the ZinQ Android app are freely available for download from Github (github.com/gtStyLab/zinQapp).

## Data availability

All data needed to evaluate the conclusions in the paper are present in the paper or the Supplementary Materials. The source data underlying Figs. 1–6, and Supplementary Figs. 2, 5, 6, 7, 8, 9, 10, and 11 are provided as a Source Data file. All other data are available upon reasonable request.

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

## Acknowledgements

We thank Julie Champion and her lab for use of their freeze-drier. This work was supported by the National Institutes of Health (R01-EB022592 and R35-GM119701) and the National Science Foundation (MCB-1254382). M.P.M. was supported by an NSF graduate research fellowship (DGE-1650044).

## Author contributions

Conceptualization: M.P.M. and M.P.S.; Investigation: M.P.M., C.L.M., and K.K.; Formal Analysis: M.P.M.; Writing – Original Draft: M.P.M.; Writing – Review & Editing: M.P.M. and M.P.S.; Visualization: M.P.M. and M.P.S.; Supervision: J.S. and M.P.S.; Funding Acquisition: M.P.S.

## Competing interests

M.P.S. is founder and officer of a company, Chromanostics, previously formed for commercialization of micronutrient diagnostics. The authors declare no other competing interests.
