## [Peer Review File · Nature Communications]

Reviewers' Comments:

Reviewer #1:

Remarks to the Author:

The authors present a system for evaluating the concentration of zinc in serum. Their goal is to provide an easy to use, field-deployable system for monitoring zinc efficiency, a public health concern. They build on previous work on sensor and reporter engineering (Watstein et al., ACS synbio, 2018). They tune the sensitivity threshold and response time of the system, and test it in serum.

The authors invested significant efforts in improving their initial zinc sensor, and should be acknowledged for that. Yet, as a general conclusion, I believe the work presented here does not present sufficient conceptual, methodological or practical advances for publication in Nature Communications.

My first comment is about the novelty and originality, and how this work goes beyond what has been previously achieved in the field. The system presented here is very close to the previous one from the authors that was published in ACS synbio (2018). More specifically, the use of a ligand responsive promoter coupled to a reporter or to an inverter driving a different output is per se not new circuit engineering. Nor is the tuning of the inverter response by changing RBSs and SsrA tags. Neither the fact to add inducer to derepress the system at the time of use. The reporter system was already optimised and characterised by the authors in previous articles. In all, novel conceptual advances in circuit design or implementation types are not demonstrated.

My second comment is about the applicability of this system. Using existing systems, improving them using known methods to make them useable in a realistic context is actually important to go beyond the proof of concept. So I'd be fine with point #1 if this was the case here. Yet, the demonstration that the system is actually usable in the field, on patient samples, and by people that would actually performs test in POC for example, is not provided here. It is unclear how the evaluation of the color would be efficiently performed by various people, especially given the faint differences between the red and yellow pigments. One question is still how such system would be used in the field, with the inherent variability in sensor behavior due to changes in temperature, handling/storage, operator, etc...The authors talk about using cell phone to take a picture, that could be then analysed in comparison with a standard, but do not implement this idea. In all, the demonstration of field applicability is lacking. Again, I acknowledge that these things are not easy to perform, but without this demo, and without the conceptual advances discussed above, my opinion is that the paper will not be of broad interest for the general audience of Nature Communications.

In conclusion, while the reviewer appreciate the efforts taken by the authors to improve their system, the general conclusion is that this work would be more appropriate for a more specialized journal.

Other comments:

- On figure 1D and sup. fig 2. Fig1D, heat map: the value of the data point in the first column, second row is unexpectedly low. I do not understand why the signal in *placZnu, 2A* (IPTG(-)/Zinc (+)) is higher than the signal in IPTG (-), Zinc (-). Especially because this construct has one less operator for LacI than *placZnu, 2B*, which has a higher basal level. It seems the bar for initial condition without any inducer is missing. Is this an error in plotting the data?
- In fig3 and 4, the *Zur* gene is actually used as a constitutive repressor. While the authors state this fact in the discussion, it is unclear why the use of *Zur* was still necessary-maybe using another repressor like CI or beta would provide a system with a better behaviour.
- it is not clearly explained why 25% serum was chosen. While this seems to stem from the

authors previous work (Watstein et al., ACS Synbio, 2018), nothing is explained here. Was the serum heat inactivated for example? did the authors used high concentration of cells to overcome problems found in here earlier work? details are lacking.

- if the authors chose to use 5 or 10% serum, maybe they would not had to tune their sensor in the first place?

- On the same line, is the presence of glucose in serum could also explain the drop in performance?

- How do the cells behave at the single cell level? are all the cells responding or only part of the population-this could be addressed by flow cytometry and GFP reporters.

- The circuit diagrams highlighting the invert would help the reader understand the circuit design at a higher level.

- "pideal" is not explained and should be placed separately from the rest. is it a simulation? or a real experiment?

- the current methods to measure zinc and other efforts to make a POC device should be discussed.

In fig 1A, label the name of pigments outputs.

fig 1B, the LacI does not control Our expression or function, so the arrows are not correctly allocated.

fig1E, labels should include inducer (IPTG) concentration

line 247 The promoter here should be PLacZnu,2B?

line 261 Figure 3C, there is no Figure 2C

line 262, Give a code name for the chosen construct , it will be easier to follow.

line 422, zur in italics

mat and methods 37°C

in Figure 3B and C, code names for constructs would help.

Reviewer #2:

Remarks to the Author:

Remarks to the Author:

The originality of this manuscript includes: (1) they built a dual-input promoter that respond to both IPTG and zinc; (2) the color switch point was rationally tuned by redesigning regulatory circuit architecture; (3) the attempt of linking biosensor output (pigments) with RGB color score. The manuscript is set up in a very illustrative way, however all utilized methods in this manuscript were already established and the biosensor architectures were used in a similar form in earlier work. I have to recognize that the amount of work put into this study is certainly impressive, in particular considering the efforts to improve the response range to physiologically relevant zinc concentrations in human serum, but this is not the novelty desired for Nat. Comms.

Major Comments:

(1) The RGB color score can be calculated from image of cell pellets. However, this quantitative color assessment of cells relies on a constant biomass. In this study, the OD was control at 3.0. Does it make a significant difference on RGB color score if you change the OD to a higher value or a lower value?

(2) To prove the specificity of this zinc biosensor, the effect of a mixed metal ions on pigment production should be tested.

(3) The detailed sequence information of many genetic parts (such as promoters, degradation tag

and RBS) used in this study is missing. The author only provides a relative strength of these genetic parts, which is far from enough.

(4) Is the data used of the analysis of the constructs publicly available? – I did not see any links to it.

Minor comments

(1) Please change “dextrose” to “glucose” to be consistent with the nomenclature in the paper (Line 455). Please change the numeric order of *ssrA* tag, because most readers will assume that *ssrA* tag with a higher value refers to a strong degradation strength similar to RBS strength.

(2) In the protein mutagenesis experiments, 37C should be 37°C.

(3) References in the manuscript and the supplementary materials should be carefully checked. There are many mistakes in this section (references should be revised in the same format; all scientific names of organisms need to be italicized...).

Response to Reviews

We thank the reviewers for the thoughtful and insightful suggestions for our manuscript. We have implemented all of the changes suggested by the reviewers, which have led to substantial changes in the manuscript. We believe that, as the reviewers suggested, the changes have significantly improved the manuscript and made it better suited for publication in *Nature Communications*. Below we address each of the comments made by the reviewers and indicate where in the manuscript we have made the suggested changes. All original reviewer comments are in blue.

Reviewer #1 (Remarks to the Author):

The authors present a system for evaluating the concentration of zinc in serum. Their goal is to provide an easy to use, field-deployable system for monitoring zinc efficiency, a public health concern. They build on previous work on sensor and reporter engineering (Watstein et al., ACS synbio, 2018). They tune the sensitivity threshold and response time of the system, and test it in serum.

The authors invested significant efforts in improving their initial zinc sensor, and should be acknowledged for that. Yet, as a general conclusion, I believe the work presented here does not present sufficient conceptual, methodological or practical advances for publication in Nature Communications.

We thank the reviewer for the feedback and appreciate that the threshold for publication in *Nature Communications* is high. We have made several substantial additions to the manuscript (based on comments below) that we believe significantly improve its quality, novelty, and impact to make it appropriate for publication in this journal. In particular, the level of field-deployability we now demonstrate (specifically in terms of sample size, equipment requirements, robustness to environmental variability, in addition the sensor-level advances we described in the first draft) is substantially beyond what is in the published literature for whole-cell sensors, showing substantial practical (and arguably methodological) advances.

My first comment is about the novelty and originality, and how this work goes beyond what has been previously achieved in the field. The system presented here is very close to the previous one from the authors that was published in ACS synbio (2018). More specifically, the use of a ligand responsive promoter coupled to a reporter or to an inverter driving a different output is per se not new circuit engineering. Nor is the tuning of the inverter response by changing RBSs and SsrA tags....

We thank the reviewer for this detailed assessment of our work here, and for clearly taking the time to read up on and assess its relationship to our previously published manuscripts. While we acknowledge that we have used similar sensing approaches in previous papers, the earlier systems did not respond to zinc concentrations near physiologically relevant levels, which is absolutely critical for in-field zinc assessment and a significant part of our claim that the original work was a major advance towards field application.

Additionally, though we presented an initial “inverter” in our previous *ACS Synthetic Biology* paper, we showed that it could only modestly shift threshold concentrations when fluorescent reporter were used, demonstrating proof-of-principle but no practical application. Because of the many complexities associated with metabolic pigment production, it was insufficient to adequately control violacein expression. Further, since the inverter setup presented in that work relied on the addition of decoy operator sites, plasmids were highly unstable, which resulted in a high degree of variability among replicates, construct instability, and inconsistent and unreliable cell pigmentation. In this manuscript, we overhaul the circuit architecture, overcoming all of those problems. We have extended our description of previous efforts at circuit tuning (lines 261-262) to better convey the limitations of our previous work and the challenges that this work overcomes.

...Neither the fact to add inducer to derepress the system at the time of use. The reporter system was already optimised and characterised by the authors in previous articles. In all, novel conceptual advances in circuit design or implementation types are not demonstrated.

While we have used inducible promoters to de-repress pigment production pathways in previous papers (we believe the reviewer alludes in part to our *Metabolic Engineering* publication), those were only used to control individual pigment pathways, and were only used for single-input control: they were not used to control response to a target molecule to be sensed. Effectively controlling multiple metabolic pathways in cells is much more challenging, and even more challenging is to control those pathways based on two separate inputs (one for repression/derepression and one for sensing), and to do so required redesigning the circuit architecture and creating new synthetic promoters to enable accurate control.

My second comment is about the applicability of this system. Using existing systems, improving them using known methods to make them useable in a realistic context is actually important to go beyond the proof of concept. So I'd be fine with point #1 if this was the case here. Yet, the demonstration that the system is actually usable in the field, on patient samples, and by people that would actually performs test in POC for example, is not provided here. It is unclear how the evaluation of the color would be efficiently performed by various people, especially given the faint differences between the red and yellow pigments. One question is still how such system would be used in the field, with the inherent variability in sensor behavior due to changes in temperature, handling/storage, operator, etc...The authors talk about using cell phone to take a picture, that could be then analysed in comparison with a standard, but do not implement this idea. In all, the demonstration of field applicability is lacking. Again, I acknowledge that these things are not easy to perform, but without this demo, and without the conceptual advances discussed above, my opinion is that the paper will not be of broad interest for the general audience of Nature Communications.

We thank the reviewer for this comment and the suggestion of concrete steps we could take to demonstrate potential in-field use. We wholeheartedly agree that demonstration of some of the capabilities raised here would have a major effect on the impact of our work and its importance. Since the time of submission, we have optimized cells so that they can be reliably used in field-

like conditions. We have added an entirely new section (lines 374 – 448) to the results that details additions to the test (listed below) that make it more field-deployable:

- (1) We show that sensor cells can be **lyophilized and stored at ambient temperature**, enabling storage and distribution that would be encountered in real applications. Based on the literature and our early experiments a few years ago, it was unclear that this would be possible, but we have successfully demonstrated it.
- (2) We show that tests can be **run in volumes of blood that can be isolated from a finger stick** (rather than the large volumes we had previously used), which makes it more realistic for people who would “actually perform test[s] in POC”, per the reviewer’s suggestion, as it means that a finger-stick is sufficient for the test rather than a venous blood draw that would require significant medical training. Again, a whole-cell test working in this way has not previously been demonstrated.
- (3) We show that **tests can be run on a “body incubator” that uses just body heat and agitation for cell growth**, demonstrating that tests can function without robust temperature control. Again, this speaks to field-deployability and the true minimum of equipment needed for the test to be run, rather than just assuming that it will work without lab-like incubation conditions.
- (4) We develop a **new standardization approach that accounts for user-to-user variation** and any potential variability in the testing environment, per the reviewer’s concern.
- (5) We validate the test’s accuracy **using serum isolated from multiple individual donors**, moving beyond the commercially acquired pooled samples and demonstrating robustness to matrix effects that are widely prevalent in biospecimens.
- (6) **We develop and validate a smartphone app that enables objective test interpretation at the site of testing**, which addresses concerns of “how the evaluation of the color would be efficiently performed by various people”, as the app makes it a largely objective process.

While we would seize the opportunity to join forces with people performing real micronutrient surveys, a true in-field trial would be a huge logistical undertaking, as finding the right people, at the right time, who are able to assess our test context of their IRB and other approvals, is incredibly difficult. Based on efforts to incorporate another one of our sensors into field tests, we have discovered that it is not widely known who is going out on a field campaign at any time (with government agencies being particularly logistically challenging to coordinate with), which makes connections and logistics of test incorporation quite difficult. So, while we regret that we were not able to include a full field trial that would demonstrate deployability, we assure the reviewer that it is not for lack of taking the suggestion seriously; it is just a major logistical undertaking that is beyond the scope of what we can do based on our connections developed to date, and we hope beyond the scope of the paper now that we have more clearly demonstrated field deployability through the revisions made here. The additional experiments and discussion that we have added to this manuscript demonstrate that upon establishing these connections and overcoming approval hurdles, our sensor cells are suitable for deployment in a field test.

In conclusion, while the reviewer appreciate the efforts taken by the authors to improve their system, the general conclusion is that this work would be more appropriate for a more specialized journal.

We thank the reviewer for all of the constructive feedback and believe that the changes that have resulted from these suggestions have significantly improved the manuscript. **Because we have addressed the main comments presented by this reviewer and demonstrated the real potential of this test to be used in a minimally equipped setting, we believe that this manuscript is now well suited for publication in a journal with the breadth and impact of *Nature Communications*.**

Other comments:

*- On figure 1D and sup. fig 2. Fig1D, heat map: the value of the data point in the first column, second row is unexpectedly low. I do not understand why the signal in *placZnu, 2A* (IPTG(-)/Zinc (+) is higher than the signal in IPTG (-), Zinc (-). Especially because this construct has one less operator for *LacI* than *placZnu, 2B*, which has a higher basal level. *t* seems the bar for initial condition without any inducer is missing. Is this an error in plotting the data?*

We thank the reviewer for pointing out this apparent contradiction and apologize for not representing this data sufficiently clearly. For the specified promoter, the average fluorescence of the IPTG(-)/Zn(-) condition is in fact lower than the fluorescence of the IPTG(-), Zn(+) condition, but the difference is not statistically significant. The standard deviation of the IPTG(-)/Zn(-) condition is high relative to the average value, likely because the values are so close to the plate reader's limit of detection. The heat map shown in Figure 1D does not capture variation, and the original axes of the plot in Figure S2 masked this variation. We have adjusted the axes in Figure S2 to make the similarity between the two conditions more apparent; the appearance in Figure 1D remains, though, due to limitations in the visualization strategy. We hope that the clarification for Figure S2 is sufficient to ameliorate the reviewer's legitimate concern here.

*- In fig3 and 4, the *Zur* gene is actually used as a constitutive repressor. While the authors state this fact in the discussion, it is unclear why the use of *Zur* was still necessary-maybe using another repressor like *CI* or *beta* would provide a system with a better behaviour.*

The reviewer makes a valid point in saying that we could have used a more well-characterized repressor to control our sensor's response, and we have added a sentence to the discussion stating the potential of such a change to improve the system's behavior (lines 473 – 476). Upon settling on the inverter circuit as the method for tuning response, we chose to continue using *Zur* because we had already extensively characterized a dual-input promoter with a *Zur* operator and demonstrated its high dynamic range in response to both *LacI* and *Zur* and its efficacy in the context of controlling pigment production. While other dual-input promoters have been reported to have similar dynamic ranges, most characterization has been done with fluorescent reporters; our previous experience, publications, and data show that characterization in fluorescent reporter space often does not translate well to pigment reporter space. As a result, staying with the same

promoter was the more prudent and reliable course of action, though the reviewer is correct about potential interchangeability should sufficiently effective dual-input promoters be identified or engineered.

- it is not clearly explained why 25% serum was chosen. While this seems to stem from the authors previous work (Watstein et al., ACS Synbio, 2018), nothing is explained here...

We acknowledge that the 25% serum selection was insufficiently justified in the first draft. The motivation was initially a combination of (as the reviewer alluded to) knowing from previous whole-cell work that we could get sensor cell survival in 25% serum, plus the fact that a 25% serum sample would be a simple dilution that could be done in the field with limited equipment (just one pipettor/dropper) and likely in a way robust to measurement error. Both of these are examples of the broader justification of trying to enable tests that meet the constraints of an in-field assay.

Our revision now includes reduced sample and culture sizes in our efforts to model field deployability (200 uL of 25% serum), which itself also warrants similar justification. To that end, the motivating factors for the selection of these parameters are to meet the requirements of (1) minimal sample processing to minimize user-introduced inaccuracies, (2) small sample volumes, and (3) sufficient culture size to produce a visible pellet. Test that consist of 200 μ L cultures and 25% serum can meet all three of these requirements, using no more than a finger-stick of blood with minimized workflow complexity yet still enabling sensor cell survival and visual detectability. We have added this justification to lines 202 – 209 of the manuscript.

...Was the serum heat inactivated for example? did the authors used high concentration of cells to overcome problems found inhere earlier work? details are lacking.

The serum was not heat inactivated, since that could not be reliably and reproducibly performed in a field setting. We used high concentrations of cells to overcome problems associated with poor cell growth in untreated serum. We have added details about the serum that was used in testing to lines 346 – 347 of the manuscript.

- if the authors chose to use 5 or 10% serum, maybe they would not had to tune their sensor in the first place?

The reviewer raises a valid point in suggesting that using a very diluted serum sample could potentially have precluded the need for some (though not all, or perhaps even most) tuning that we performed. However, this would require a field-tester to perform precise dilutions in the field, which would introduce variability and test inaccuracies. In our discussions with potential end users and those who supervise those potential end users, it was repeatedly conveyed to us that simplifying the sample workflow is critical to deployability, and that anything requiring significant precision will mitigate potential use and impact of a tool. To minimize necessary dilutions, while also minimizing the volume of serum needed, we aimed for tests to work in 25% serum samples. Lines 202 – 209 further detail our justification for using a 25% serum sample.

- On the same line, is the presence of glucose in serum could also explain the drop in performance?

While we did not extensively investigate the effects of serum components on test performance, it is certainly possible that glucose could alter coloration, especially since we have shown that the carbon source in the growth media strongly affects pigment production (Figure 4, Figure S9). Since we have little control over the “matrix effects” introduced by serum, we instead implemented a robust way to account for them, which is a significant addition to this paper that resulted from the reviewer’s above comments. The final section of the results details an approach for developing customized calibration curves from cells that are run in serum “standards” that could be provided as lyophilized samples, which enables the test to reliably assess zinc status regardless of the confounding effects of serum components (such as glucose) on pigment production.

- How do the cells behave at the single cell level? are all the cells responding or only part of the population-this could be addressed by flow cytometry and GFP reporters.

The reviewer poses an interesting question; because we focused on the final color of the cells, we did not investigate single-cell behavior, which could be interesting to investigate separately. As the reviewer suggested, we could use flow cytometry to assess changes in fluorescence. However, it is worth noting that fluorescent reporters do not capture the critical complications and phenomena of metabolite reporters, and so accurate single-cell characterization would require single-cell analysis of pigmentation, which would be difficult to do. To that end, we have not pursued this line of inquiry here, though we appreciate the suggestion and will investigate developing an approach moving forward to address it.

- The circuit diagrams highlighting the invert would help the reader understand the circuit design at a higher level.

Figure 3A contains a circuit diagram that details the design and intended functionality of the Zur inverter circuit; we have adjusted the caption for Figure 3A to more clearly highlight what aspects of the circuit serve as an effective “inverter” in the system.

- “pideal” is not explained and should be placed a sepaeraely from the rest. is it a simulation? or a real experiment?

We thank the reviewer for drawing this point of confusion to our attention. We included the hypothetical fluorescence values of a promoter that shows ideal behavior to make the heat map more interpretable. We have adjusted the caption of Figure 1D to more explicitly indicate that the data plotted alongside P_{ideal} represents example data. We have also restructured the plot so that it is displayed separately from the other promoters that show experimental data, per the reviewer’s suggestion.

- the current methods to measure zinc and other efforts to make a POC device should be discussed.

We have included a more detailed description of current methods to assess zinc deficiency in the Introduction (lines 68 – 73).

In fig 1A, label the name of pigments outputs.

We have added labels both to the figure and to the figure caption.

fig 1B, the LacI does not control Our expression or function, so the arrows are not correctly allocated.

We thank the reviewer for pointing out that our effort to distinguish between the pre-culture and sensing stages resulted in a representation of regulation that could give the wrong impression of what is physically occurring in the system. We have adjusted the figure and the figure caption to indicate that the circuit is equally controlled by Zur and by LacI, and we have also made the corresponding changes to the schematics in Supplementary Figure 1.

fig1E, labels should include inducer (IPTG) concentration

We have added concentrations of both IPTG and arabinose to the figure caption.

line 247 The promoter here should be PLacZnu,2B?

We thank the reviewer for catching this typo; we have made this change (now in line 268) and made the same change for multiple other instances (lines 266, 276).

line 261 Figure 3C, there is no Figure 2C

We apologize for this typo and have corrected it (now in line 281).

line 262, Give a code name for the chosen construct , it will be easier to follow.

We thank the reviewer for this excellent idea; we have named the construct in accordance with the labels now presented in Figure 3C and refer to it in the text accordingly.

line 422, zur in italics

We have made this change (now in line 660).

mat and methods 37°C

We have made this change throughout the Materials and Methods section.

in Figure 3B and C, code names for constructs would help.

We have added plasmid names to Figures 3B and 3C, which are referenced in the Figure caption and throughout the text. We have provided similar names and references for the plasmids presented in Figure 4 and in Supplementary Figure 9.

Reviewer #2 (Remarks to the Author):

Remarks to the Author:

The originality of this manuscript includes: (1) they built a dual-input promoter that respond to both IPTG and zinc; (2) the color switch point was rationally tuned by redesigning regulatory circuit architecture; (3) the attempt of linking biosensor output (pigments) with RGB color score. The manuscript is set up in a very illustrative way, however all utilized methods in this manuscript were already established and the biosensor architectures were used in a similar form in earlier work. I have to recognize that the amount of work put into this study is certainly impressive, in particular considering the efforts to improve the response range to physiologically relevant zinc concentrations in human serum, but this is not the novelty desired for Nat. Comms.

We thank the reviewer for the assessment of our manuscript and for appreciating the effort that was required to properly tune the sensor cells to respond to clinically relevant zinc concentrations. Since receiving these reviews, we have made substantial changes to the manuscript, primarily to demonstrate that the developed sensor cells can be easily used and reliably interpreted in minimally equipped settings. The demonstration that we provide for the field-deployability of these whole-cell sensors is the first of its kind. We refer the reviewer to the details provided in response to Reviewer #1's critique, but in case that review is not being made available, we would like to summarize the contributions to field deployability quickly:

- (1) We show that sensor cells can be **lyophilized and stored at ambient temperature**, enabling storage and distribution that would be encountered in real applications.
- (2) We show that tests can be **run in volumes of blood that can be isolated from a finger stick** (rather than the large volumes we had previously used), which makes it more realistic for people who would “actually perform test[s] in POC”.
- (3) We show that **tests can be run on a “body incubator” that uses just body heat and agitation for cell growth**, demonstrating that tests can function without robust temperature control.
- (4) We develop a **new standardization approach that accounts for user-to-user variation** and any potential variability in the testing environment.
- (5) We validate the test's accuracy **using serum isolated from multiple individual donors**, demonstrating robustness to matrix effects that are widely prevalent in biospecimens.
- (6) **We develop and validate a smartphone app that enables objective test interpretation at the site of testing.**

Each of these contributions is a substantial step that has not previously been demonstrated for whole-cell diagnostic tests. We believe these changes have dramatically improved the quality and impact of the manuscript, with these new implementations providing sufficient novelty to make the work appropriate for publication in *Nature Communications*.

Major Comments:

The RGB color score can be calculated from image of cell pellets. However, this quantitative color assessment of cells relies on a constant biomass. In this study, the OD

was control at 3.0. Does it make a significant difference on RGB color score if you change the OD to a higher value or a lower value?

The reviewer raises a good point in stating that cell density can affect the color score, a phenomena that we observed early in test development. While we initially used an OD of 3.0 because it gave the most visibly apparent color changes, we have since developed a standardization approach that accounts for potential variation in cell density. Briefly, we run “standards” with known concentrations of zinc in parallel with the “test” reaction and create custom calibration curves for each test based on the standards, which could be distributed in lyophilized form (this method is detailed in lines 410 – 422). While this method was primarily used as a way to account for user-to-user variation, it has the added benefit of accounting for any potential variability caused by differential cell lyophilization density or cell growth during culture.

(2) To prove the specificity of this zinc biosensor, the effect of a mixed metal ions on pigment production should be tested.

The reviewer’s point that the specificity of the sensor for zinc should be demonstrated is well-taken. We have added a figure (Supplementary Figure 10) showing that the transcription factor used in the final sensor cells does not respond to other divalent cations that are present in serum, and that the sensor still responds to zinc in the presence of these cations.

(3) The detailed sequence information of many genetic parts (such as promoters, degradation tag and RBS) used in this study is missing. The author only provides a relative strength of these genetic parts, which is far from enough.

We apologize for not including this information in the initial submission. We have added extensive tables to the Supplementary Materials that provide sequences for all of the genetic parts that were used in this study (Supplementary Tables 3 – 5)

(4) Is the data used of the analysis of the constructs publicly available? – I did not see any links to it.

We have included all data in the main manuscript or in the Supplemental Information, and we have added sequence information for the constructs used. Further, we have added a statement to the end of the manuscript detailing that any additional data or information about the data is available upon request.

Minor comments

(1) Please change “dextrose” to “glucose” to be consistent with the nomenclature in the paper (Line 455).

We have made this change (now in line 552).

*Please change the numeric order of *ssrA* tag, because most readers will assume that *ssrA* tag with a higher value refers to a strong degradation strength similar to RBS strength.*

We have made this adjustment in Figure 3, Figure 4, Figure S8, and Figure S9, and we have adjusted figure captions and in-text references appropriately.

(2) In the protein mutagenesis experiments, 37C should be 37oC.

We apologize for this oversight and have made the appropriate changes throughout the Materials and Methods section.

(3) References in the manuscript and the supplementary materials should be carefully checked. There are many mistakes in this section (references should be revised in the same format; all scientific names of organisms need to be italicized...).

We apologize for these mistakes. We have updated all citations so that organism names are properly italicized and have attempted to unify formatting.

Reviewers' Comments:

Reviewer #1:

Remarks to the Author:

The authors have provided significant improvements to the manuscript and addressed my comments. They have made their point stronger in the introduction, highlighted where the manuscript novelty was, and compared with existing methods. Experimental choices have been clarified. The field testing methods, with cell lyophilisation and phone app clearly add to the interest of the paper for a general audience and for the application of these devices in the real world. The paper is now in my opinion suitable for publication in Nat Comm.

Reviewer #2:

Remarks to the Author:

The authors have responded to all comments adequately, and I recommend the study for publication.